# Recombinant Hemagglutinin Protein from H9N2 Avian Influenza Virus Exerts Good Immune Effects in Mice

**DOI:** 10.3390/microorganisms12081552

**Published:** 2024-07-29

**Authors:** Xiaofeng Li, Zhixun Xie, You Wei, Meng Li, Minxiu Zhang, Sisi Luo, Liji Xie

**Affiliations:** 1GuangXi Key Laboratory of Veterinary Biotechnology, GuangXi Veterinary Research Institute, Nanning 530000, China; lixiaofeng2003@126.com (X.L.); weiyou0909@163.com (Y.W.); mengli4836@163.com (M.L.); 2004-luosisi@163.com (S.L.); xie3120371@163.com (L.X.); 2Key Laboratory of China (GuangXi)-ASEAN Cross-Border Animal Disease Prevention and Control, Ministry of Agriculture and Rural Affairs of China, Nanning 530000, China

**Keywords:** AIVs, recombinant, HA protein, stability, immune effects, cytokines

## Abstract

The H9N2 subtype of avian influenza virus (AIV) causes enormous economic losses and poses a significant threat to public health; the development of vaccines against avian influenza is ongoing. To study the immunogenicity of hemagglutinin (HA) protein, we constructed a recombinant pET-32a-HA plasmid, induced HA protein expression with isopropyl β-D-1-thiogalactopyranoside (IPTG), verified it by SDS–PAGE and Western blotting, and determined the sensitivity of the recombinant protein to acid and heat. Subsequently, mice were immunized with the purified HA protein, and the immunization effect was evaluated according to the hemagglutination inhibition (HI) titer, serum IgG antibody titer, and cytokine secretion level of the mice. The results showed that the molecular weight of the HA protein was approximately 84 kDa, and the protein existed in both soluble and insoluble forms; in addition, the HA protein exhibited good acid and thermal stability, the HI antibody titer reached 6 log2–8 log2, and the IgG-binding antibody titer was 1:1,000,000. Moreover, the levels of IL-2, IL-4, and IL-5 in the immunized mouse spleen cells were significantly increased compared with those in the control group. However, the levels of IL-1β, IL-6, IL-13, IFN-γ, IL-18, TNF-α, and GM-CSF were decreased in the immunized group. The recombinant HA protein utilized in this study exhibited good stability and exerted beneficial immune effects, providing a theoretical basis for further research on influenza vaccines.

## 1. Introduction

Avian influenza viruses (AIVs) are members of the *Orthomyxoviridae* family and can infect a wide variety of hosts, including birds, pigs, dogs, horses, bats, and humans [1,2,3,4]. According to differences in virion envelope surface hemagglutinin (HA) proteins and neuraminidase (NA) antigens, AIVs can harbor H1-H16 and N1-N9 [5], which are combined into a variety of virus subtypes, all of which endanger the development of the poultry industry. In China, the H9N2 AIV was first reported in Guangdong Province in 1992; subsequently, the disease spread to other provinces, where it became an endemic strain. Due to genetic evolution, the pathogenicity and transmissibility of the virus have shown increasing trends annually, thus posing serious threats to the poultry industry. In addition, H9N2 can not only directly infect humans but also recombine with other subtypes of viruses (H5N1, H7N9, H10N8, and H5N6) to produce variant subtypes of viruses that threaten human health [6,7,8,9,10].

In general, local farming and trade patterns provide a means for the spread of H9N2 epidemics. Traditional small-scale free-range and polyculture systems, which account for a portion of national poultry production, have low vaccination coverage and biosecurity awareness, and freshly killed poultry meat is preferred over frozen meat in live poultry markets (LPMs). These methods have significantly contributed to a vast gene pool of AIVs, as evidenced by the increased prevalence of viruses, including multiple HA/NA subtypes [11].

Currently, H9N2 AIV is widespread in China and has become a stable strain on commercial flocks [12]. However, antigenic variation or drift persists, putting pressure on the compatibility of seasonal vaccines. Mutations in the HA protein loci (R164Q, T150A, and I220T) promote virus replication in both poultry and mammals. The molecular mechanism of the antigenic transformation of avian influenza A subtype H9N2 has been elucidated, providing an important reference for the selection of vaccine candidates [13]. Since 2013, H9N2 circulating in Chinese poultry has undergone frequent mutations at HA residue 193, and viruses carrying asparagine (N) have been replaced by viruses carrying alanine (A), aspartic acid (D), glutamic acid (E), glycine (G), and serine (S). These mutations greatly alter the antigenicity of H9N2 and have profound impacts on its replication and transmission in chickens [14]. Moreover, the H9N2 AIV is divided into two antigenic clades from the original H9.4.2.5 lineage, with new clades emerging after 2015 and old clades reappearing in other regions in 2018. Hemagglutinin antigenic drift has led to the emergence of new clades, and vaccine immunity is a challenge [15]. A survey has shown that H9N2 is the predominant AIV subtype in wholesale and retail markets, mainly in the forms of G57 and three new genotypes (NG164, NG165, and NG166), and the emergence of NG165 genotypes, which are more suitable for transmission among poultry and mammals, poses a threat to public health [16]. In one case, H9N2 influenza viruses isolated from chickens, humans, and pet cats were found to have a high degree of homology, suggesting that H9N2 AIV may coexist in poultry and mammals, thus threatening both human and animal health [17]. Although the mortality rate of H9N2-infected flocks is usually no more than 20%, an H9N2 infection usually leads to respiratory and egg drop symptoms as well as severe secondary infections with other respiratory diseases, thus affecting poultry productivity [18]. The continuous antigenic mutation of the H9N2 subtype of avian influenza virus has challenged the prevention and control effects of vaccines, and research and development into immune mechanisms and new vaccines are also urgently needed.

Inactivated vaccines are currently the main vaccines available for influenza viruses, and they rely on embryo production. Following immunization, humoral immunity predominates, and the abilities to induce effective mucosal and cellular immunity are lacking. The effectiveness of a vaccine is limited by differences in genetic variants and subtypes of the virus. Therefore, the development of new, safe, and highly effective avian influenza vaccine candidates is critical, and new DIVA vaccines against H9N2 have emerged, including recombinant live virus vector vaccines, subunit vaccines, DNA vaccines, and virus-like particle vaccines. Subunit vaccines are generally developed based on the extraction of AIV immunogenic proteins rather than on the introduction of viral particles.

As the major viral membrane protein, HA is the primary protective antigen used in current influenza vaccines due to its immunodominant induction of high neutralizing antibody titers. Studies have shown that the combination of HA and NA mRNAs from different influenza subtypes can induce potent antibody and cellular immune responses in immunized non-human primates [19]. The HA protein of AIVs has rich epitopes and exhibits strong immunogenicity; therefore, this protein stimulates the host to produce neutralizing antibodies and has become the main target protein in the development of influenza vaccines. The insertion of HA gene fragments into different viral vectors can result in a strong resistance to host virus infections [20,21,22]. Some studies have also explored the preparation of HA stems and glycosylation sites in different groups of influenza viruses for use in vaccines to improve cross-antibiotic protection [23]. Research has shown that H5 COBRA HA vaccines induce protective antibodies against two historical H5Nx and currently circulating H5N1, H5N6, and H5N8 clade influenza viruses [24]. Deng et al. [25] expressed the HA of the H1 and H3 strains and the M2e sequences common to humans, pigs, and poultry to form a bilayer nanovaccine, which achieved good immune effects in immunized mice. The abovementioned studies showed that the HA protein exerts a good immune effect, but the cytokines produced via cellular immunity remain to be explored.

In the present study, we cloned the H9N2 HA protein and constructed plasmid pET-32a-HA. The acid resistance and thermal stability of the HA protein were determined after purification. The HA protein was prepared for the immunization of BALB/c mice, and the immune effect was evaluated based on the resulting antibody and cytokine levels. Our results suggest that vaccination with the HA protein induces an immune response, increases or decreases cytokine expression, and prepares the host for protection without significant side effects, which have important implications for the development of new antiviral vaccines.

## 2. Materials and Methods

### 2.1. Ethics Statement

The animal experiments and sample collection were conducted in accordance with the guidelines of protocol #2019C0409, issued by the Animal Ethics Committee of the Guangxi Veterinary Research Institute.

### 2.2. Virus and Mouse Strains

The A/chicken/Guangxi/CWM/2019 (H9N2) virus strain used in this study was isolated from the livers of diseased chickens in Nanning, Guangxi, and stored at −70 °C at the Guangxi Veterinary Research Institute. Our laboratory studied this strain and found that it has potential use for a vaccine. Six- to eight-week-old female BALB/c mice (Guangdong Animal Center, Foshan, China) were maintained in cages with independent air supply.

### 2.3. HA Gene Encoding and Plasmid Construction

Based on the HA sequence of the A/chicken/Guangxi/CWM/2019 (H9N2) strain, we amplified the full-length coding sequences (CDS) of the HA gene, and the amino acids from positions 1 to 560 encoded by the full-length CDS were expressed, referring to the program (https://web.expasy.org/translate/, accessed on 24 July 2024). The primers (HA-U and HA-D) were designed (Table 1) and synthesized by Guangzhou Ruibo Biotech (Guangzhou, China). The HA gene was cloned from cDNA obtained from reverse transcription of RNA extracted from the virus and used as a template, and the PCR products were separated by 1.5% agarose gel electrophoresis; then, the PCR product was recovered and inserted into the pMD18-T plasmid (Takara Bio, Dalian, China, Code No. 6011) and transferred into competent DH5α cells, where the plasmid pMD18-T-HA was constructed. The pET-32a(+) plasmids and pMD18-T-HA plasmids were digested with BamH Ⅰ (Takara Bio, Code No. 1010A) and Not Ⅰ (Takara Bio, Code No. 1166A) at 37 °C for 4 h; then, the target fragments were recovered and ligated overnight at 16 °C. Finally, the ligated product was transfected into competent BL21 cells, which were verified by sequencing by Ruibo Biotech Company (Guangzhou, China). Since the pET-32a(+) vector carries two His tags, when the HA gene fragment with the start codon and the stop codon was inserted into the plasmid, the HA protein was expressed and fused with one of the His tags on the vector.

### 2.4. SDS–PAGE and Western Blotting

HA protein expression was induced with 1.0 mmol/L isopropyl β-D-1-thiogalactopyranoside (IPTG, Beyotime Biotechnology, Beijing, China, Code No. I1020) for 6 h. The bacteria were centrifuged at 4000 rpm for 5 min and then washed three times with 1× PBS (Solarbio, Beijing, China, Code No. P1020). The cells were then lysed with RIPA lysis buffer (Solarbio, Code No. R0010) on ice for 30 min and subjected to ultrasonication at 50 Hz until thoroughly lysed. The cells were boiled for 10 min and then incubated on ice for 10 min. The suspension (total protein), supernatant (soluble protein), and precipitate (insoluble protein) were collected after centrifugation for SDS–PAGE (Solarbio, Code No. PG01010). The recombinant proteins were stained with Coomassie brilliant blue (Beyotime Biotechnology, Code No. P1300). If the HA protein was present in the supernatant, it was considered a soluble protein; otherwise, it was considered an insoluble protein. After SDS–PAGE separation, the proteins were transferred to PVDF membranes, and the membranes were incubated with Western blot blocking solution for 4 h. The membranes were incubated with a diluted anti-His mouse monoclonal antibody (Invitrogen, Carlsbad, CA, USA, Code No. 26183). After the primary antibody was removed, the membranes were washed three times (10 min each time) with 1× PBST buffer (Solarbio, Code No. 1033), incubated with HRP-labeled goat anti-mouse IgG (Invitrogen, Code No. C31430100) for 1 h, and washed four times with 1× PBST buffer. Photographs were taken after color development using a 3,3′-diaminobenzidine (DAB) horseradish peroxidase color development kit (Beyotime Biotechnology, Code No. P0202).

### 2.5. Purification HA Protein

After HA protein expression for 6 h, the bacteria were centrifuged at 8000 rpm for 5 min and then washed three times with 1× PBS, lysed with lysis buffer (CW BIO, Beijing, China, Code No. CW0894S) on ice for 30 min, and then lysed via ultrasonication at 50 Hz until the cells were thoroughly lysed. The cells were then centrifuged at 10,000 rpm for 10 min to collect the supernatant, and the supernatant was mixed with binding buffer (CW BIO, Code No. CW0894S). The mixture was loaded onto a Ni-Agarose resin column (CW BIO, Code No. CW0894S), and the columns were mixed on a horizontal shaker for 30 min. The protein was completely bound to the resin. After elution of impurities with 15 volumes of binding buffer, the HA protein was eluted with different concentrations (100, 120, 160, 170, 180, 190, 200, 210 mM) of imidazole elution buffer (CW BIO, Code No. CW0894S), and the flow-through solution was collected. After purification, the protein was dialyzed with PBS to completely dialyze the imidazole of the eluate. SDS–PAGE and Western blot were also performed to verify that the obtained proteins was HA protein. Aliquots of purified HA protein were stored at −70 °C.

### 2.6. Hemagglutination Assay of HA Protein

To test the effect of the HA purified protein (antigen) on red blood cells (antibodies), the purified HA protein was tested with 1% chicken red blood cells (prepared and provided by the Guangxi Key Laboratory of Veterinary Biotechnology) to determine whether it caused red blood cell aggregation. Normal saline (25 μL, Solarbio, Code No. IN9000) was added to wells 1–12 of the coagulation plate, HA protein (25 μL) was added to the first well and mixed well, and 25 μL of the mixture from the first well was added to the second well. The 10th well was diluted in this manner. That is, the purified HA protein solution was diluted 1:2, 1:4, 1:8, 1:16, 1:32, 1:64, 1:128, 1:256, 1:512, and 1:1024, where the last two wells were for red blood cell control, and the other row of wells were for saline control. Finally, 25 μL of 1% red blood cells was added to wells 1–12, which were gently vortexed and mixed. The results were observed after manual mixing and incubation at 37 °C for 30 min. The experiment was repeated 3 times.

### 2.7. HA Protein Acid Sensitivity

The HA purified protein (200 µg/mL, 100 μL) was mixed with equal volumes of 100 mmol/L (pH 4.0, pH 5.0) acetate buffer (Yuanye Bio-Technology, Shanghai, China, Code No. R26128, R26131), pH 6.0 phosphate buffer (Yuanye Bio-Technology, Code No. 26268), and pH 7.0 neutral phosphate buffer (Yuanye Bio-Technology, Code No. 26273) and incubated at 37 °C for 10 min, after which the change in stability was determined using a hemagglutination assay.

### 2.8. HA Protein Thermal Sensitivity

The HA purified protein (100 µg/mL, 100 μL) was incubated in a water bath maintained at 56 °C for durations of 0, 5, 10, 15, 30, or 60 min, after which its stability was determined via a hemagglutination assay. The HA protein was incubated at 56 °C for 5 min, after which the coagulation stability decreased by 2 unit titers, indicating thermal instability. The protein was then incubated at 56 °C for 30 min. If the hemagglutination titer decreased by 2 unit titers, the protein was considered thermostable. Otherwise, it was considered to be moderately thermally stable, exhibiting properties between those of thermostable and moderately thermally stable proteins.

### 2.9. Animal Experiments and Safety Evaluation

All animal studies were conducted in accordance with the recommendations of the Guide for the Care and Use of Laboratory Animals of the Ministry of Science and Technology of the People’s Republic of China as well as with the institutional guidelines of the Animal Ethics Committee of Guangxi Veterinary Research Institute. All animal experiments and sample collections were conducted in accordance with the guidance set forth in protocol #2019C0409, which was issued by the Animal Ethics Committee of the Guangxi Veterinary Research Institute.

Six- to eight-week-old female BALB/c mice were randomly assigned to the two following groups (*n* = 10) for our study: the HA protein-immunized group and the Freund’s adjuvant-immunized group (mock). The HA protein (100 µg/mL) was thoroughly mixed with complete Freund’s adjuvant (Sigma-Aldrich, Shanghai, China, Code No. 1003150981) and emulsified to a uniform consistency. Intraperitoneal solution at a dose of 0.5 mL was applied on a 14-day schedule. For the second immunization, the complete adjuvant was replaced with incomplete Freund’s adjuvant (Sigma-Aldrich, Code No. 1003212978), and the immunization methods were identical. The Freund’s adjuvant-immunized group was injected with an equal volume of adjuvant and PBS. The mental state of the mice was monitored on a daily basis. Following each immunization, the mice were weighed, and their deaths were recorded over a 14-day period. The results are presented as the mean body weight of five mice.

### 2.10. Hemagglutination Inhibition Antibody Assays

Following the administration of a booster immunization to the mice at 14 days, blood was obtained retro-orbitally, and sera were treated with receptor-destroying enzyme (RDE, Denka Seiken, Japan, Code No. 340016) at 56 °C for 30 min. Serial dilutions of the mouse sera (25 μL) were mixed with 4 units of H9N2 viral fluid and incubated at 37 °C for 30 min. Fresh chicken red blood cells were shaken, and 50 μL of the cell suspension was pipetted into each well. The mixture was thoroughly mixed and allowed to stand at room temperature for 30 min to observe the results. The test was conducted using serum from each mouse.

### 2.11. IgG Antibody Assay

Mouse serum was collected as described above. The purified HA protein (10 µg/mL) was diluted with 1× ELISA coating buffer (Solarbio, Code No. C1055), which was used as the antigen, and added to a 96-well ELISA plate (100 µL/well). The plate was stored at 4 °C overnight and then washed 3 times with 1× PBST buffer. The plate was blocked with 1% bovine serum albumin (Solarbio, Code No. 7940) at 37 °C for 1 h. The plate was washed as described above; mouse serum was diluted with PBS at dilutions of 1:10,000, 1:20,000, 1:30,000, 1:40,000, 1:50,000, 1:60,000, 1:70,000, 1:80,000, 1:90,000, 1:100,000, 1:150,000, and 1:200,000 (each concentration was replicated in 3 wells), and negative mouse serum or PBS was used as the control. The plate was washed 3 times with 1× PBST, incubated with goat anti-mouse IgG (Proteintech, Wuhan, China, Code No. SA00001-1) at 37 °C for 1 h, and washed 3 times with 1× PBST. The OD450 was measured using a VICTOR^®^ Nivo™ Multimode Plate Reader (PerkinElmer) after color development using chromogenic kit (Solarbio, Code No. PR1200) and stop solution (Solarbio, Code No. C1058), and the maximum dilution of the positive group OD value (P)/negative group OD value (N) > 2 represented the mouse serum IgG antibody titer.

### 2.12. Mouse Splenocyte Supernatant Multicytokine Assay

Following the removal of the spleens from the sacrificed mice and thorough homogenization, the cells were incubated at room temperature for five minutes and then resuspended in red blood cell lysis buffer (Solarbio, Code No. R1010). Lysis was terminated with Dulbecco’s modified Eagle’s medium (DMEM; Gibco, Code No. C11885500BT). The cells were washed 3 times with PBS, resuspended in supplemented DMEM, supplemented with 5% fetal bovine serum (FBS; Gibco, Grand Island, NY, USA, Code No. 10099141C1), distributed into 6-well plates (1.0 × 10^6^ cells/well), and cultured at 37 °C in 5% CO_2_ and 95% humidity. Subsequently, the cells in the HA protein-immunized group were divided into two groups, designated as HA+ and HA−, while cells from the mock group were divided into two groups, designated as mock+ and mock-. The HA+ and mock+ groups were subjected to stimulation with the HA protein, whereas the cells in the HA− and mock− groups were not stimulated. After 72 h, the cell supernatant was collected and sent to the Shanghai Leids Biotechnology Company for cytokine detection. The experiment was repeated three times, and results were calculated as follows, IL-2 (HA) = mean (HA+) − mean(HA−), IL-2 (mock) = mean (mock+) − mean (mock−).

### 2.13. Statistical Analysis

The statistical analyses were conducted using t-tests with GraphPad Prism 8.0 software. Asterisks indicate statistical significance as follows: * indicates *p* < 0.05; ** indicates *p* < 0.01.

## 3. Results

### 3.1. HA Protein Successfully Expressed in Escherichia coli Cells

The HA gene was amplified using cDNA as the PCR template, and cDNA was reverse transcribed from RNA extracted from viruses. The sequence analysis revealed that the size of the HA gene was 1683 bp (Figure 1A). The SDS–PAGE results showed that the HA protein was present in both the supernatant and precipitate, indicating that it existed in both soluble and insoluble protein forms (Figure 1B). The size of the HA protein with His-tag was approximately 84 kDa; meanwhile, using an anti-His mouse monoclonal antibody to verify the HA protein, a band of approximately 84 kDa was confirmed for the recombinant plasmid pET-32a-HA, but the corresponding band was not observed for the empty vector (Figure 1C). When the HA protein mixture was purified, the SDS–PAGE results showed that the 180 mM imidazole eluate eluted best (Figure 1D) and highly concentrated and pure proteins can be obtained. Meanwhile, the Western blot result showed that the obtained purified protein was the HA protein (Figure 1E).

### 3.2. HA Protein Has Good Acid and Thermal Stability

The results show that the HA protein can cause hemagglutination on red blood cells at a dilution of 1:128 (Figure 2A).The hemagglutination titer of the HA protein was 7 log2. No hemagglutination of red blood cells occurred in the red blood cells (well C) and saline control group. When the purified HA protein was treated with different pH buffer solutions and different temperatures, the same method was used to experiment and determine the results.

The acid sensitivity of the HA protein was measured using hemagglutination titers after incubation with solutions at different pH values. First, the HA protein was incubated with neutral phosphate buffer (pH 7.0), acetate buffer (pH 5.0), and acetate buffer (pH 4.0), and the hemagglutination titers were 7 log2, 6 log2, and 4.7 log2, respectively (Figure 2A). The stability of the HA protein was good in the neutral solution, and the stability decreased rapidly in a more acidic environment. The thermal stability of the HA protein changed with increasing incubation time. The HA proteins were incubated at 56 °C for 5 min or 15 min, and the titer decreased from 6 log2 to 5 log2, indicating that the HA protein had good stability in this temperature range. When the HA protein was incubated for 30 min or 60 min, the titers decreased to 4.7 log2 and 4.3 log2, respectively (Figure 2B), representing a decrease in the titer of 2 units from the preincubation hemagglutination titer and indicating that the recombinant HA protein had good stability; these results show that the stability of the HA protein decreased with increasing incubation time.

### 3.3. The HA Protein Is Safe in Mice

The safety of immunizing mice with the recombinant HA protein was evaluated according to the weight change and survival rate. After immunization with HA recombinant protein, the change in the body weight of the mice was greater than that of the control group; the mice showed a decrease in body weight and then returned to their original body weight at either the first (HA-1) or second (HA-2) immunization, and they showed an increase in body weight over time compared with those in the control group (Figure 3A). All mice survived after the first (Mock-1) and second (Mock-2) immunizations (Figure 3B), suggesting that the purified HA protein is relatively safe for use in mice.

### 3.4. High Levels of Hemagglutinin-Inhibiting Antibodies and IgG Antibodies

The immunogenicity of the HA protein was determined by measuring antigen-specific humoral immune responses after boost immunizations. An HI antibody titer assay was performed with serum from immunized mice. After 2 weeks of immunization, specific antibodies against the H9 subtype of AIV were produced in the serum of the immunized mice (Figure 4A). The HI antibody titers of HA protein-immunized mice reached 6 log2–8 log2, and the antibody titers of the control group were negative.

To determine the level of IgG antibody produced in mice after immunization, the HA protein was used as the antigen, and mouse serum was used as the antibody. The ELISA results showed that the IgG titer produced by the immunized mice was 1:1,000,000 (Figure 4B), which was significantly different from that of the control group (*p* < 0.01). The results indicated that HA protein-immunized mice produced specific IgG-binding antibodies.

### 3.5. Effects of Multiple Cytokines on Mouse Spleen Cell Culture

The above results suggest that the recombinant HA protein produces antibodies after immunization in mice; however, it is unclear whether cytokines are involved in the immune response. To explore the changes in cytokine levels, we collected mouse splenocyte supernatants for cytokine assays. Cytokine concentrations in the mouse splenocyte culture supernatant were determined using Luminex. The levels of IFN-γ, IL-2, IL-4, IL-5, TNF-α, IL-1β, IL-6, IL-13, IL-18, and GM-CSF were measured (Figure 5). Compared with those in the control group, the levels of IL-2 (Figure 5A), IL-4 (Figure 5B), and IL-5 (Figure 5C) in the cells of the immunized group were significantly increased (*p* < 0.01); among them, the levels of IL-2 increased the most, and the average level in the control group was 5.15 pg/mL, which increased to 60.48 pg/mL after immunization. However, the levels of IL-1β, IL-13, IL-18, IFN-γ, and GM-CSF were significantly decreased (*p*< 0.01). In particular, the average level of IL-18 in the immune group (7191.07 pg/mL) was significantly lower than that in the control group (21,596.6 pg/mL). In addition, the levels of IL-6 and TNF-α were lower in the immunized group than in the control group, but the difference was not significant (*p* > 0.05). These results suggest that the HA protein is involved in antiviral immunity, increasing the levels of IL-2, IL-4, and IL-5 or decreasing the levels of other cytokines.

## 4. Discussion

The HA protein is the major antigenic protein on the surface of AIV and is the antigen of choice for subunit avian influenza vaccines. Recombinant antigenic proteins are produced by genetic engineering, in which genes with multiple antigenic sites are cloned into expression vectors and protein expression is induced to yield a single sexual antigen, a well-established technique that produces antigens with a favorable safety profile for both humans and animals.

In this study, the HA protein was expressed by prokaryotes and existed in both soluble and insoluble protein forms. The HA protein was incubated in a buffer at pH values of 4.0, 5.0, 6.0, or 7.0 for 10 min, and the stability of the HA protein decreased with decreasing pH. In addition, when the HA protein was mixed with a buffer (pH 6.0) and acidic buffer (pH 4.0), the hemagglutination titer of the protein began to decrease by approximately 2 units. The reason for this difference may be that the prokaryotically expressed proteins are unstable under acidic conditions due to inadequate structural folding and may degrade. In addition, the thermal stability measurements showed that the titer of the HA protein changed slightly after incubation at 56 °C for 5 min to 15 min, suggesting that the stability of the HA protein was strong. After incubation at 56 °C for 30 min, the titer decreased to 4.7 log2, indicating that the thermal stability of the HA protein was good. These results indicate that the recombinant HA protein has good thermal stability; therefore, it may also have good thermal stability when prepared as a vaccine.

Subunit vaccines induce humoral and cellular immune responses following the immunization of the host. In the present study, after the mice were immunized with the HA protein, their weight first decreased and then increased, and all mice survived, indicating that the recombinant HA protein is safe for mice. After two weeks, the HI antibody level reached 6 log2–8 log2, and the IgG antibody titer was 1:1,000,000, indicating that the recombinant HA protein induced a good immune response in mice and that this antibody can promote the neutralization of the virus. T cell-mediated cellular immunity is now a popular area of vaccine research, including as a direction in the production of general influenza vaccines [26,27,28,29]. The influenza vaccine induces a cellular immune response that can provide broad-spectrum protection; multiple immune responses contribute to the prevention of an influenza infection. Among these responses, antibodies alone provide adequate protection against an infection, and T cell-mediated responses appear to play an important role in recovery [30]. After the mice were immunized with HA in our study, the concentrations of IL-2, IL-4, and IL-5 in the supernatant were significantly higher than those in the mock group, and the levels of IL-2 and IL-4 were consistent with the results reported by Feifei Xiong et al. [31]. IL-2, IL-4, and IL-5, which are mainly secreted by cells, stimulate innate immunity and modulate host immune responses [32,33]; therefore, the secretion of these cytokines indicates that the immunization of mice elicits a strong cellular response.

Compared with those in the control group, the levels of IL-1β, IL-13, IFN-γ, IL-18, and GM-CSF were significantly reduced in the cells of the immunized group (*p* < 0.01), and the levels of IL-6 and TNF-α were reduced, but the differences were not significant (*p* > 0.05).These results suggest that compared with the control mice injected with a mixture of Freund’s adjuvant and PBS, in the mice immunized with the HA protein and Freund’s adjuvant, the treatment causes an increase or decrease in cytokine levels and protects the host through different pathways [31,34,35]; thus, the immunization results showed that HA had a good immune effect. The production of inflammatory factors has dual effects; at a certain concentration, inflammatory factors can inhibit viral replication, but excessive accumulation can lead to host inflammatory damage and acute death. The cytokines IL-1β and IL-18 induce an inflammatory response after an influenza virus infection and recruit other inflammatory cells to infected tissues to clear the virus. Levels of IL-1β and IL-18 were significantly lower in the immune group than in the control group (*p* < 0.01). In the early stage of a viral infection, IL-1β and IL-18 promote the activity of CD8+ T cells and induce antibody secretion, thus playing protective roles; however, when overproduced, these cytokines destroy benign tissues [36]. IL-1β and IL-18 bind to receptors and induce NF-κβ-dependent inflammation [35]. IL-18 can mediate IFN-γ production via T and NK cells [37]. IFN-γ is a primary antiviral agent, but when mass production leads to adverse outcomes, IFN-γ has been shown to play an important role in acute lung injury caused by an H1N1 virus infection [38]. Some studies have shown that when the host is infected with H1N1, H3N2, or H7N9, excessive IL-1β can exacerbate the condition and cause serious consequences. In both the early and late stages of H1N1 or H3N2 infections, treatment with targeted anti-IL-1β antibodies can alleviate lung inflammation and favorably improve survival [36,39]. In our study, IL-1β levels decreased after the immunization of mice with HA, indicating that recombinant HA can alleviate lung inflammation in mice by reducing the secretion of inflammatory cytokines. IL-1β and IL-18 play complex roles in the influenza virus-induced cytokine storm. These cytokines not only play important roles themselves but also regulate the production of TNF-α and IL-6. Compared to the control group, the levels of IL-5, IFN-γ, and GM-CSF in the immunized group were significantly lower (*p* < 0.05), while the levels of IL-6 and TNF-α were not significantly different (*p* > 0.05). IL-6 is an inflammatory marker, and clinical studies have shown that excess IL-6 in people with influenza is associated with adverse effects [40,41,42]. Compared to those in WT mice, IL-6 levels in SOCS3-/- mice infected with influenza virus are significantly lower and return to normal levels, preventing the production of cytokine storms [43]. IL-6 plays an important role in the cytokine storm caused by influenza viruses and represents a novel target for immunotherapeutic strategies. TNF-α is a typical proinflammatory factor located at the center of the cytokine storm [44]. Like IFN, TNF-α can disrupt the endothelial barrier, causing pulmonary edema and tissue damage [45]. Compared to WT mice, mice without TNFR are more resistant to lethal H5N1, have an average survival time of two days, and exhibit lower levels of cytokines, including IFN-γ and interleukins, in their lungs [46,47]. Therefore, controlling TNF-α levels to reduce cytokine storms while inhibiting viral replication is a potential strategy through which to reduce pathologic damage in the lungs. Both inflammatory factors can protect the host from a viral infection; conversely, they can also exacerbate inflammatory damage to the host. The results described above indicate that the recombinant HA protein can not only produce IgG antibodies but also promote the secretion of cytokines and reduce the accumulation of some inflammatory factors, which may involve different methods of protecting the host.

## 5. Conclusions

In conclusion, the H9N2 recombinant HA protein induces cellular and humoral immune responses, results that lay the foundation for an in-depth study of the function of the HA protein in immunity and vaccination; these findings also provide a reference for the effective use of viral proteins to inhibit viral replication as well as insights for the further research and development of influenza vaccines.

## Figures and Tables

**Figure 1 microorganisms-12-01552-f001:**
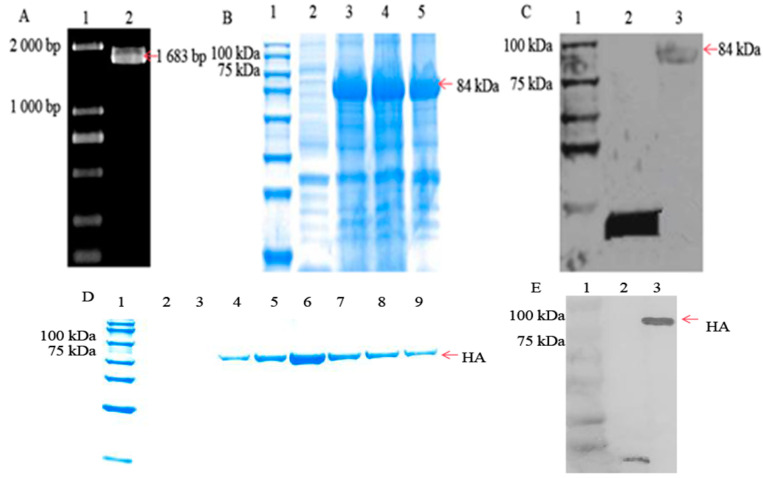
The analysis of the HA gene and recombinant HA protein. (**A**) The PCR product of the *H9 HA* gene. Lane 1: DL 2000 DNA marker; Lane 2: PCR amplification products. The red arrow indicates the amplified HA fragment, the size of which is 1683 bp. (**B**) The SDS–PAGE and solubility analysis of the recombinant protein. Lane 1: protein marker; Lane 2: protein obtained from the empty pET-32a plasmid; Lane 3: protein obtained from the pET-32a-HA plasmid (total protein); Lane 4: protein from the supernatant (soluble protein); Lane 5: protein from the precipitation (insoluble protein). The red arrow indicates the recombinant HA protein, which is 84 kDa, and the His-tag protein is included. (**C**) The identification of recombinant HA via Western blotting. Lane 1: protein marker; Lane 2: protein obtained from the pET-32a empty plasmid; Lane 3: protein obtained from the pET-32a-HA plasmid. The red arrow indicates the recombinant HA protein, which is 84 kDa, and the His-tag protein is included. (**D**) The SDS‒PAGE of HA purified protein. Lane 1: protein marker; Lanes 2–9: Proteins eluted with 100, 120, 160, 170, 180, 190, 200, and 210 mM imidazole eluent; 180 mM imidazole eluted best. The red arrow indicates the HA purified protein. (**E**) The identification of HA purified protein via Western blotting. Lane 1: protein marker; Lane 2: protein obtained from the pET-32a empty plasmid; Lane 3: His tag HA purified protein. The red arrow indicates the HA purified protein.

**Figure 2 microorganisms-12-01552-f002:**
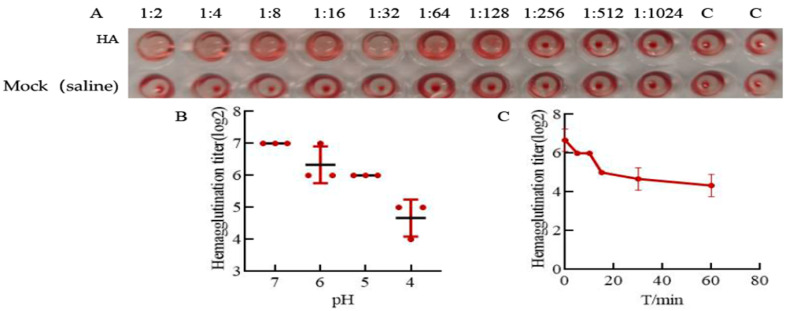
Acid stability and thermal stability results. (**A**) The result of HA protein hemagglutination. HA protein hemagglutination at 1:2, 1:4, 1:8, 1:16, 1:32, 1:64, and 1:128 dilutions of red blood cells. The hemagglutination titer of the HA protein was 7 log2. There is no hemagglutination of red blood cells that occurred in the red blood cells (well C) and saline control group. (**B**) The acid stability results. The HA protein was incubated with a neutral phosphate buffer (pH 7.0), phosphate buffer (pH 6.0), and acetate buffer (pH 4.0 or 5.0), and the hemagglutination titers were 7 log2, 6.3 log2, 6 log2, and 4.7 log2, respectively. (**C**) The result of thermal stability. The HA protein was incubated in a 56 °C water bath for 0, 5, 10, 15, 30, or 60 min, and the hemagglutination titers were 6.7 log2, 6 log2, 6 log2, 5 log2, 4.7 log2, and 4.3 log2, respectively. These hemagglutination assay results were determined through log2 calculations.

**Figure 3 microorganisms-12-01552-f003:**
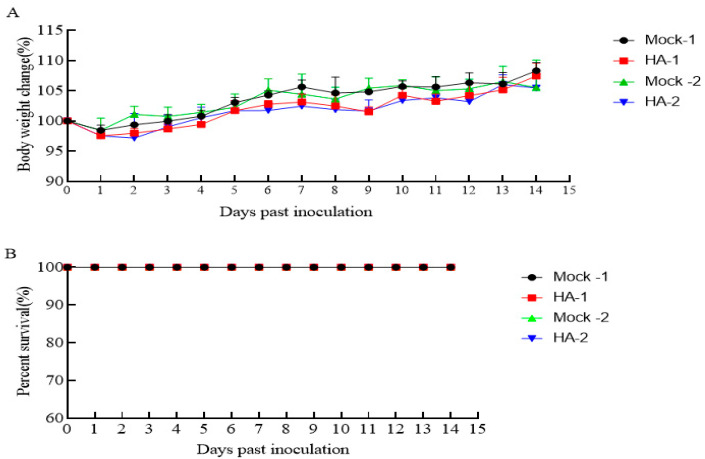
The safety of the recombinant HA protein in mice. Following each immunization, the body weight and survival of the mice were monitored for 14 consecutive days. (**A**) Changes in the body weight of mice. The results are shown as the average body weight of 5 mice. Mock-1, HA-1 indicates a change in body weight after the first immunization; Mock-2, HA-2 indicates a change in body weight after the second immunization. (**B**) Percent survival (%). There were no deaths after the first and second immunizations. Mock-1, HA-1 indicates the first immunization, Mock-2, HA-2 indicates the second immunization.

**Figure 4 microorganisms-12-01552-f004:**
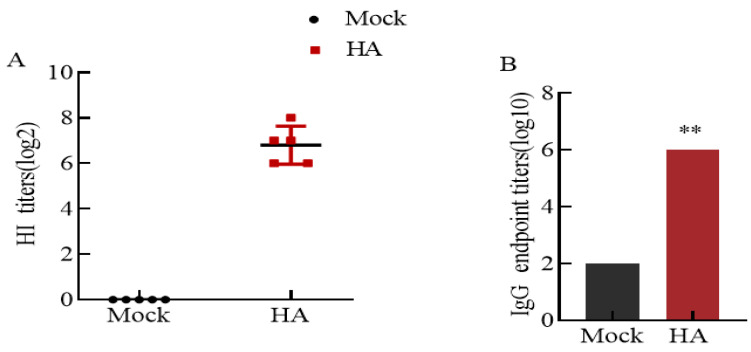
The results of the HI antibody and IgG antibody. (**A**) The results of the HI antibody. The serum of the control group mice did not show anticoagulation in the HI experiment, and the HI antibody titer was negative. The HI antibody titer of HA protein-immunized mice reached 6 log2–8 log2. (**B**) The IgG antibody results. The results showed that HA protein-immunized mice produced specific IgG-binding antibodies. Asterisks indicate significant differences (** indicates *p* < 0.01).

**Figure 5 microorganisms-12-01552-f005:**
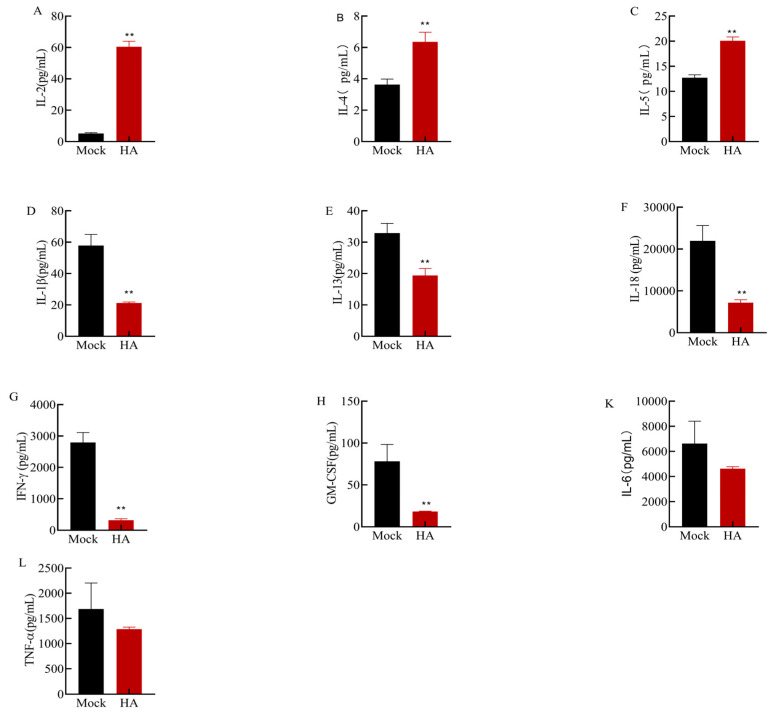
Luminex assay results for mouse splenocyte culture supernatants. (**A**) IL-2 levels in both groups were 5.15 pg/mL and 60.48 pg/mL, respectively. (**B**) IL-4 levels in both groups were 3.64 pg/mL and 6.36 pg/mL, respectively. (**C**) IL-5 levels in both groups were 12.74 pg/mL and 20.10 pg/mL, respectively. (**D**) IL-1β levels in both groups were 57.84 pg/mL and 21.25 pg/mL, respectively. (**E**) IL-13 levels in both groups were 32.92 pg/mL and 19.37 pg/mL, respectively. (**F**) IL-18 levels in both groups were 21,956.6 pg/mL and 7191.07 pg/mL, respectively. (**G**) IFN-γ levels in both groups were 2794.4 pg/mL and 318.7 pg/mL, respectively. (**H**) GM-CSF levels in both groups were 78.08 pg/mL and 18.35 pg/mL, respectively. (**K**) IL-6 levels in both groups were 6633.1 pg/mL and 4621 pg/mL, respectively. (**L**) TNF-α levels in both groups were 1688.03 pg/mL and 1286.08 pg/mL, respectively. Asterisks indicate significant differences (** indicates *p* < 0.01).

**Table 1 microorganisms-12-01552-t001:** Primers used in this study.

Primer Name	Sequence (5′-3′)	Amplified Sequence Length
HA-U	ATAGGATCCATGGAGACAGTATCACTAATAACTA	1683 bp
HA-D	ATAAGAATGCGGCCGCTTATATACAAATGTTGCATCTG	

## Data Availability

The original contributions presented in this study are included in the article; any further inquiries can be directed to the corresponding author.

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
