# Peer review of "Recombinant Hemagglutinin Protein from H9N2 Avian Influenza Virus Exerts Good Immune Effects in Mice"

_microorganisms, 2024, doi:10.3390/microorganisms12081552_

Round 1

Reviewer 1 Report

Comments and Suggestions for Authors

The authors expressed HA protein of H9N2 virus in E. coli and purified. They tested stability of the HA and immunogenicity in mice. They saw high HI and binding IgG antibody using big amount of HA protein mixed with Freund’s adjuvant. The biggest problem of the mouse experiments is absence of proper control group. The authors should have used Freund’d adjuvant alone as a control.

Major points:

1.     Line 120: Please indicate amino acid numbers expressed in E. coli.

2.     Line 170: Please indicate the concentration of HA protein used.

3.     Please include photo of HA assay with endpoint concentration for hemagglutination.

4.     Figure 3: To show the safety, you need to include data from the first immunization and control group should receive vehicle injection rather than unimmunized.

5.     Lines 399-401: This argument can not apply to HA protein as it is not an internal antigen.

6.     Line 413: Without comparing with Freund’s adjuvant-injected mice, you can’t say these are HA derived-cytokines. I think what you saw are caused by Freund’s adjuvant.

Minor points:

1.     Line 83: Please delete “hemagglutinin”.

2.     Line 131: Please delete “/min”.

3.     Line 248: Please delete “approximately”.

4.     Figure 1B: Please amend the position of 100 and 75 kDa.

Author Response

Response to reviewer 1

Major points:

  1. Line 120: Please indicate amino acid numbers expressed in E. coli.

Response:Thank you for pointing this out. We agree with this comment. Therefore,

we answer as follows:

“The CDS coding region of the HA gene sequence is 1,683 bp, encoding a total of 560 amino acids.” And add this to the Line 125-126 of the manuscript. Please refer to the manuscript for details.

  1. Line 170: Please indicate the concentration of HA protein used.

Response:Thank you for pointing this out. We agree with this comment. Therefore,

we answer as follows:

In our HA protein acid sensitivity assay, the concentration of HA purified protein used was 200 µg/mL(Line 191) mixed with equal volumes of buffers of different pH,to achieve a final concentration of 100 µg/mL. For the HA protein thermal sensitivity assay, the concentration of HA purified protein used was 100 µg/mL(Line 196), and these protein concentrations are reported in the manuscript.

  1. 3.Please include photo of HA assay with endpoint concentration for hemagglutination.

Response:Thank you for pointing this out. We agree with this comment. Therefore,

we answer as follows:

The purified HA protein solution was diluted 1:2, 1:4, 1:8, 1:16, 1:32, 1:64, 1:128, 1:256, 1:512, 1:1024,with the last two wells were for red blood cell control, and the other row of wells were for saline control, and reacted with red blood cells. Finally, 25 μL of 1% red blood cells were added to wells 1-12, which were gently vortexed and mixed. The mixture was incubated for 30 min at room temperature(Line 184-186). The results show that HA protein can cause hemagglutination on red blood cells at a dilution of 1:128(Figure 2A).The hemagglutination titer of HA protein was 7 log2 . There is no hemagglutination of red blood cells occurred in the red blood cells( well C) and saling control grous. When the purified HA protein was treated with different pH buffer solutions and different temperatures,the same method was used to experiment and determine the results.(Line 301-306)

Figure 2 (A) The result of HA Protein Hemagglutination. HA protein Hemagglutination at 1:2, 1:4, 1:8, 1:16, 1:32, 1:64, 1:128 dilutions of red blood cells .The hemagglutination titer of HA protein was 7 log2 .  There is no hemagglutination of red blood cells occurred in the red blood cells( well C) and saling control grous.(Line 325-329).

4  Figure 3: To show the safety, you need to include data from the first immunization and control group should receive vehicle injection rather than unimmunized.

Response:Thank you for pointing this out. We agree with this comment. Therefore,

we answer as follows:

Our mouse experiments were conducted were conducted as follows, six-to eight-week-old female mice were randomly assigned to the following two groups (n=10) for our study: the HA protein-immunized group and the Freund's adjuvant -immunized  group(Mock,the original manuscript of the Line197-198, present manuscript of the Line219-220). HA protein-immunized group: purified HA protein (100 µg/mL) emulsified with complete Freund's adjuvant (Sigma), administered a solution intraperitoneally, at a dose of 0.5 mL. Freund's adjuvant -immunized group(mock):PBS emulsified with complete Freund's adjuvant (Sigma) in a multipoint injection, at a dose of 0.5 mL. The mental state of the mice was monitored on a daily basis. After the first immunization, the mice were weighed and their deaths were recorded over a 14-day period. For the second immunization after 14 days, the complete adjuvant was replaced with incomplete Freund's adjuvant (Sigma), and the immunization methods were identical. And the same,the mental state of the mice was monitored on a daily basis. The mice were weighed and their deaths were recorded over a 14-day period. The results are presented as the mean body weight of five mice.

Our control group was injected with a mixture of PBS and Freund's adjuvanthe (The original manuscript of the Line197-198, present manuscript of the Line219-220) , since the purified HA protein contained an imidazole eluate, which would be dialyzed out with PBS buffer before immunizing the mice, and the final purified HA protein was dissolved in PBS. So control group used PBS instead of HA protein.

Your suggestion of injecting the vector in the control group is very much appreciated as a very good suggestion, and we will take this into account in our future experiments.

After immunization with HA recombinant protein, the change in body weight of the mice was greater than that of the control group, the mice showed a decrease in body weight and then returned to their original body weight at either the first or second immunization, and showed an increase in body weight over time, compared with those in the control group (Figure 3A). All mice survived after the first and second immunizations(Figure 3B). suggesting that the purified HA protein is relatively safe for use in mice.

The details can be found in Line 337-343 of the manuscript.

The weight changes and survival rates of the mice after the first and second immunization are as follows, added to the results of the experiment.

Figure 3  Safety of recombinant HA protein in mice. Following each immunization, the body weight and survival of the mice were monitored for 14 consecutive days. (A) Changes in body weight of mice. The results are shown as the average body weight of 5 mice. Mock-1, HA-1 indicates change in body weight after first immunization; Mock-2, HA-2 indicates change in body weight after second immunization (B) Percent survival (%). There were no deaths after first and second immunizations. Mock-1, HA-1 indicate first immunization,Mock-2, HA-2 indicate second immunization.(Line 346-352)

  1. Lines 399-401: This argument can not apply to HA protein as it is not an internal antigen.

Response: Thank you for pointing this out. We agree with this comment. Therefore, we answer as follows:

 The argument here is not appropriate enough, after we think about it, remove this sentence“Specific cellular immune responses, directed against conserved internal antigens of influenza viruses, are widely recognized as the primary mechanism mediating cross-protection against influenza infection”.

  1. Line 413: Without comparing with Freund’s adjuvant-injected mice, you can’t say these are HA derived-cytokines. I think what you saw are caused by Freund’s adjuvant.

Response: Thank you for pointing this out. We agree with this comment. Therefore,we answer as follows:

The formulation here is not rigorous enough, we modify this formulation as follows:” These results suggest that, compared with control mice injected with a mixture of Freund's adjuvant and PBS, mice immunized with HA protein and Freund's adjuvant cause an increase or decrease in cytokine levels and protectthe host through different pathways; thus, the immunization results showed that HA  had a good immune effect”.

The details can be found in Line 443-447 of the manuscript.

Minor points:

  1. 1.Line 83: Please delete “hemagglutinin”.

Response:Thank you for pointing this out. We agree with this comment.

 Line83 "Hemagglutinin" was deleted. Please refer to the manuscript for details.

  1. Line 131: Please delete “/min”.

Response: Thank you for pointing this out. We agree with this comment.

Line 131  "/min" was deleted. Please refer to the manuscript for details.

  1. Line 248: Please delete “approximately”.

Response: Thank you for pointing this out. We agree with this comment.

Line 248 “approximately”was deleted. Please refer to the manuscript for details.

  1. Figure 1B: Please amend the position of 100 and 75 kDa.

Response: Thank you for pointing this out. We agree with this comment.

The position of 100 and 75 kDa has been modified in Figure 1B. Specific details can be seen in the manuscript and in the figure below.

Reviewer 2 Report

Comments and Suggestions for Authors

Li et al. reported recombinant protein expression and analysis of HA from H9N2 in E. coli. The rHA protein stability and immunogenicity were evaluated. Influenza vaccine development is a sparsely researched topic, and Li et al's study does not introduce a novel approach or significantly contribute to vaccine development, but it does provide information on the stability and immunogenicity of rHA from H9N2.

The introduction is interestingly written and includes sufficient information on the latest developments in influenza vaccine development. Probably, it is good at the very beginning of the introduction to compare the pathogenicity of H9N2 with the other AIV strains.

L29  Orthomyxoviridae – all Latin names need to be in Italic

L120 Please, explain the method of insertion of the His tag  

2.5. Purification and Hemagglutination Assay of HA Protein, split this paragraph into two points, and move the  Purification of HA before SDS PAGE and Western blot

Please, provide the pictures of SDS PAGE and Western blot of His tag  HA purified protein.

Please specify if you have done the acid and thermal stability studies with purified protein.

The discussion should be rewritten, concentrating from the beginning on a discussion of your results. L347-373 belongs more to the introduction.

Many punctuation and grammatical errors are visible in the text.

Comments on the Quality of English Language

Many punctuation and grammatical errors are visible in the text.

Author Response

1  L29  Orthomyxoviridae – all Latin names need to be in Italic

Response: Thank you for pointing this out. We agree with this comment.

“Orthomyxoviridae” has been changed to italic”Orthomyxoviridae”, The details can be found in Line 29.

2  L120 Please, explain the method of insertion of the His tag  

Response: Thank you for pointing this out. We agree with this comment.

 we carefully considered your comment and provide details below:

   The HA gene was cloned from cDNA obtained by reverse transcription of RNA extracted from the virus and used as a template, and the PCR products were separated by 1.5% agarose gel electrophoresis, then the PCR product was recovered and inserted into the pMD18-T plasmid (Takara Bio, China), and transferred into competent DH5α cells, the plasmid pMD18-T-HA was constructed. The pET-32a(+) plasmids and pMD18-T-HA plasmids were digested with BamHâ… (Takara Bio) and Notâ… (Takara Bio) at 37℃ for 4 h, then the target fragments were recovered and ligated overnight at 16℃. Finally, the ligated product was transfected into competent BL21 cells, which was verified by sequencing by Ruibo Biotech Company. Since the pET-32a(+) vector carries two His tags, when the HA gene fragment with the start codon and the stop codon was inserted into the plasmid, the HA protein was expressed and fused with one of the His tags on the vector.

The details can be found in manuscripts Line127-138.

3  2.5. Purification and Hemagglutination Assay of HA Protein, split this paragraph into two points, and move the  Purification of HA before SDS PAGE and Western blot

Response: Thank you for pointing this out. We agree with this comment.

The original manuscript of the 2.5 “Purification and Hemagglutination Assay of HA Protein”, has been split into two points. The Details can be found in manuscripts Line162 (2.5), 177 (2.6)

The experimental procedure is as follows, after the recombinant protein is expressed, it is verified by SDS-PAGE and Western blot to see whether it is HA protein, and then it is purified. After purification, it will be further verified by Western blot, and the subsequent acid and heat stability tests and mouse tests will be performed, therefore, we recommend that the purification of HA protein be performed after SDS-PAGE and Western blot.

4  Please, provide the pictures of SDS‒ PAGE and Western blot of His tag  HA purified protein.

Response: Thank you for pointing this out. We agree with this comment.Therefore,

we answer as follows:

The following figure shows the SDS‒PAGE and Western blot of the purified His-tag HA protein, and Figure 1D and E are also added to the manuscript.

“When the HA protein mixture was purified, the SDS‒PAGE results showed that, the 180 mM imidazole eluate eluted best (Figure 1D), highly concentrated and pure proteins can be obtained. Meanwhile, the Western blot result showed that the obtained purified protein was HA protein(Figure 1E). “The above is added to the manuscript, which can be found in Line 276-280.

Figure 1 (D) SDS‒ PAGE of the  HA purified protein. Lane 1: protein markerï¼›Lane2-9:Proteins eluted with 100, 120, 160, 170, 180, 190, 200, 210 mM imidazole eluent, 180 mM imidazole eluted best. The red arrow indicates the purified HA protein.(E) Identification of HA purified protein via Western blotting. Lane 1: protein marker; Lane 2: protein obtained from the pET-32a empty plasmid; Lane 3: His tag  HA purified protein. The red arrow indicates the HA purified protein.(Line 293-298)

5  Please specify if you have done the acid and thermal stability studies with purified protein.

Response: Thank you for pointing this out. We agree with this comment.Therefore,

we answer as follows:

 The acid(Line 191)and thermal stability(Line 196)in this manuscript were performed after the HA protein was purified. As this paper mainly focuses on the study of HA proteins, the unpurified HA proteins have many heteroproteins, and the thermal and acid stability of different proteins may be different, so we carried out the acid and thermal stability study on the HA proteins after purification.

6  The discussion should be rewritten, concentrating from the beginning on a discussion of your results. L347-373 belongs more to the introduction.

Response:Thank you for pointing this out. We agree with this comment.Therefore,

after careful consideration, we have rewritten the L347-373 as follows

After careful consideration, we have rewritten the L347-373 as follows

“The HA protein is the major antigenic protein on the surface of AIV and is the antigen of choice for subunit avian influenza vaccines. Recombinant antigenic proteins are produced by genetic engineering, in which genes with multiple antigenic sites are cloned into expression vectors and protein expression is induced to yield a single sexual antigen, a well-established technique that produces antigens with a favorable safety profile for both humans and animals.”The details can be found in Line 401-406.

Round 2

Reviewer 1 Report

Comments and Suggestions for Authors

1.     Line 112: Please include Catalogue numbers where applicable so that the reader can follow your methods.

2.     Line 125: Please indicate amino acid numbers expressed in E. coli. I asked amino acid position you have expressed. Does 560 mean from amino acid 1 to 560?

3.     Line 225: You can’t freeze blood. Have you treated sera with RDE? How much volume of sera and HA did you use?

4.     Lines 303 & 327: “saling” should be “saline”.

5.     Line 444: Please add references to support your argument that “increase or decrease in cytokine levels and protect the host”.

Comments on the Quality of English Language

There are several repetion. e.g. Line 187. "after manual shaking, mixing". Please edit the manuscript carefully.

Author Response

  1.   Line 112: Please include Catalogue numbers where applicable so that the reader can follow your methods.

Response:Thank you for pointing this out. We agree with this comment.

We have added the catalog numbers in Line 112 "Materials and Methods" and the details are as follows

pMD18-T plasmid (Takara Bio, China, Code No. 6011),Line132

BamHâ…  (Takara Bio,Code No. 1010A) and Not â… (Takara Bio, Code No. 1166A),Line135

IPTG (Beyotime Biotechnology, Beijing, China,Code No.  I1020 ), Line145

PBS (Solarbio, Beijing, China,Code No. P1020 ),Line 147

RIPA lysis buffer (Solarbio, Code No. R0010),Line 149

SDS‒PAGE(Solarbio,Code No. PG01010),Line 153

Coomassie brilliant blue (Beyotime Biotechnology, Code No. P1300 ), Line 155

anti-His mouse monoclonal antibody (Invitrogen, Carlsbad, CA, USA, Code No. 26183), Line 160

PBST buffer (Solarbio, Code No. 1033),Line 161

HRP-labeled goat anti-mouse IgG (Invitrogen,Code No. C31430100 ),Line 162

3,3'-diaminobenzidine (DAB) horseradish peroxidase color development kit (Beyotime Biotechnology,Code No. P0202 ),Line 165

lysis buffer (CW BIO, Beijing, China,Code No. CW0894S), Line 169

binding buffer (CW BIO,Code No. CW0894S ),Line172

Ni-Agarose resin column (CW BIO,Code No. CW0894S ),Line173

imidazole  elution buffer (CW BIO, Code No. CW0894S),Line 177

normal saline ( Solarbio, Code No. IN9000 ),Line 186

(pH 4.0, pH 5.0) acetate buffer (yuanye Bio-Technology, Shanghai, Code No. R26128, R26131), pH 6.0 phosphate buffer (yuanye Bio-Technology, Code No. 26268 ), and pH 7.0 neutral phosphate buffer (yuanye Bio-Technology, Code No. 26273), Line 196-198

complete Freund’s adjuvant (Sigma-Aldrich, Shanghai, China,Code No. 1003150981),

Line 222

incomplete Freund’s adjuvant (Sigma-Aldrich, Code No. 1003212978), Line 225

receptor destroying enzyme ( RDE, Denka Seiken, Japan, Code No. 340016),Line 233

ELISA coating buffer (Solarbio,Code No. C1055),Line 241

bovine serum albumin (Solarbio,Code No. 7940),Line 244

goat anti-mouse IgG (Proteintech,Wuhan, China, Code No. SA00001-1),Line 249

chromogenic kit (Solarbio, Code No. PR1200),Line 252

stop solution (Solarbio,Code No. C1058 ),Line 253

red blood cell lysis buffer (Solarbio, Code No. R1010), Line258

Dulbecco’s modified Eagle’s medium (DMEM; Gibco, Code No. C11885500BT), Line260

fetal bovine serum (FBS; Gibco, Grand Island, NY, USA, Code No. 10099141C1), Line262

  1. Line 125: Please indicate amino acid numbers expressed in E. coli. I asked amino acid position you have expressed. Does 560 mean from amino acid 1 to 560?

Response:Yes, we have amplified the full length CDS of the HA gene, the amino acids from positions 1 to 560 encoded by the full-length CDS were expressed. For details, please refer to the program (https://web.expasy.org/translate/), and can be found in the manuscript Line 126-127.

  1. Line 225: You can’t freeze blood. Have you treated sera with RDE? How much volume of sera and HA did you use?

Response:Thank you for pointing this out.

Yes.We took mouse serum and treated sera with RDE(56℃ for 30 min). We used unfrozen blood for the “Hemagglutination Inhibition Antibody Assays”.

Details can be found in the manuscript Line 229-230.

For the "Hemagglutination Inhibition Antibody Assays", the volume of sera is 25 μL (Line 234).

For the “Hemagglutination Assay of HA Protein”the volume of sera and HA is 25 μL(Line 186)

  1. Lines 303 & 327: “saling” should be “saline”.

Response:Thank you for pointing this out. We agree with this comment.

We have corrected lines 303 and 327: “saling” to “saline”. Details can be found in the manuscript Lines 313 & 337.

  1. Line 444: Please add references to support your argument that “increase or decrease in cytokine levels and protect the host”.

Response:Thank you for pointing this out. We agree with this comment.

We have added references(31,34,35) that support the argument that "increase or decrease cytokine levels and protect the host". This can be seen in the manuscript on Line 455.

  1.  Xiong, F.; Zhang, C.; Shang, B.; Zheng, M.; Wang, Q.; Ding, Y.; Luo, J.; Li, X. An mRNA-based broad-spectrum vaccine candidate confers cross-protection against  heterosubtypic influenza a viruses. Emerg Microbes Infect2023, 12, 2256422.(Line598)

  1.  Kong, X.; Lu, X.; Wang, S.; Hao, J.; Guo, D.; Wu, H.; Jiang, Y.; Sun, Y.; Wang, J.; Zhang, G.; Cai, Z. Type I interferon/STAT1 signaling regulates UBE2M-mediated antiviral innate immunity in a negative feedback manner. Cell reports (Cambridge)2023, 42, 112002.(Line 604)
  2. Ren, H.; Wang, S.; Xie, Z.; Wan, L.; Xie, L.; Luo, S.; Li, M.; Xie, Z.; Fan, Q.; Zeng, T. et al. Analysis of Chicken IFITM3 Gene Expression and Its Effect on Avian Reovirus Replication. Viruses2024, 16, 330, doi:10.3390/v16030330.(Line 607)

Comments on the Quality of English Language

There are several repetion. e.g. Line 187. "after manual shaking, mixing". Please edit the manuscript carefully.

Response:Thank you for pointing this out. We agree with this comment.

We have removed "shaking".Details can be found in the manuscript Line 192.

We read the manuscript carefully and corrected the punctuation and spelling errors found in the manuscript.

Reviewer 2 Report

Comments and Suggestions for Authors

I agree with the author's comments and changes. I see quite a few punctuation errors that need to be fixed.

Author Response

I agree with the author's comments and changes. I see quite a few punctuation errors that need to be fixed.

Response:Thank you for pointing this out. We agree with this comment.

We read the manuscript carefully and corrected the punctuation and spelling errors found in the manuscript.
